# Engaging undergraduate medical and dental students with academic medicine: The Aberdeen INSPIRE summer school

**Valerie Speirs**\* , **Rasha Abu-Eid, Phyo Kyaw Myint**

School of Medicine, Medical Sciences and Nutrition, University of Aberdeen, Aberdeen, United Kingdom

\* valerie.speirs@abdn.ac.uk

## Abstract

UK medical and dental school curricula limit opportunities for students to gain experience in research. This parallels a decline in the number of clinical academics. To address this at grass roots level, we organised and arranged a residential summer taster week; INSPIRE (**I**ntroducing **N**ew **S**kills to **P**romote **I**nspirational **R**esearch **E**xperience)-Aberdeen). The purpose was to give first and second year medical and dental students who wished to explore a potential clinical academic career a taste of wet laboratory research and to gain experience in basic research skills. Seventeen students from eight different UK medical and dental schools attended this free residential course and were exposed to various laboratory techniques with clinical translation and application in diagnostic and therapeutic medicine. Students were given access to relevant online learning tools of the techniques being used beforehand and seminar style presentations were used to emphasise their clinical application. Students met daily with clinical academics from different specialities to give them a flavour of potential clinical academic career pathways and options. All students felt that the summer school helped them consider academic medicine as a career thus achieving our aim to inspire clinical academics of the future.

## Introduction

Medical and dental school teaching is moving away from the traditional way of learning scientific theory in the formative years before moving to clinical settings. Newer approaches include patient-centric problem-based learning and case- or enquiry-based learning styles [1]. While some medical and dental schools still engage in traditional teaching, these are in the minority, meaning student exposure to scientific theory is lost for most students. Moreover, the focus of most medical/dental schools nowadays is to train future doctors and dentists to meet the needs of an ageing population with increasingly complex multimorbidity burden. This change in focus parallels the decline of recruitment into academic medicine internationally [2–4]. Consequently, medical/dental students are often unaware of an academic route as a career choice with few pursuing this. This is a concern; every clinical intervention is rooted in laboratory science and our future healthcare professionals should be cognisant of the principles of evidence-based medicine. Engaging students, at grass roots level, with research at the early career stage

**Data Availability Statement:** All relevant data are within the paper.

**Funding:** This study was financially supported by The Academy of Medical Sciences (https://

acmedsci.ac.uk/inspire-round-5) in the form of an INSPIRE Round 5 grant (IR5\1043) received by VS, RAE, and PKM. This study was also financially supported by Aberdeen Clinical Academic Training Programme (https://www.abdn.ac.uk/smmsn/acat/) in the form of an award received by PKM. This study was also financially supported by Cyril and Margaret Gates Charitable Trust (Aberdeen, UK) in the form of an award received by VS. The funders had no role in study design, data collection and analysis, decision to publish, or preparation of the manuscript.

**Competing interests:** The authors have declared that no competing interests exist.

is one way of addressing this decline. We report the outcome of INSPIRE (**I**ntroducing **N**ew **S**kills to **P**romote **I**nspirational **R**esearch **E**xperience)-Aberdeen, a free one-week residential summer school which ran from 25–29 July 2022 and aimed to introduce first- and second-year UK medical and dental students to laboratory research to help them consider a future career as a clinical academic. This was run by experienced investigators from different fields of medical science and designed to give students practical experience of basic laboratory skills as well as opportunities to engage with clinical academics from various medical/dental specialities throughout the weeklong course.

## Materials and methods

The summer school was advertised through the INSPIRE Leads of the Academy of Medical Sciences INSPIRE Programme, UK Medical and Dental School Councils, and the personal contacts of the organisers at UK medical and dental schools. Participants were selected through a competitive application process. Travel and subsistence were covered, and bursaries provided to offset travel costs. Prior to attending participants were given access to relevant online learning tools to familiarise themselves with the techniques that they would use these beforehand and were instructed to download relevant software packages required for data analyses. Clinical academics at various stages of training, including those in PhD programmes up to senior consultants from different disciplines (geriatric medicine, breast surgery, oral surgery, gastroenterology, rheumatology, oncology, pathology, orthodontics) gave informal talks/panel discussions arranged over 4 sessions. Evening social activities included 2 informal dinners; one with medical students from the University of Aberdeen and another with the organisers and clinical academics who participated in the programme.

On the first day students were introduced to laboratory safety followed by practising pipetting and loading gels. Four practical sessions followed on subsequent days, covering cell culture, histology & digital pathology, flow cytometry and polymerase chain reaction (PCR). Seminar style introductory presentations were used to emphasise the clinical application of the techniques used. The programme was fixed with all students completing the same activities.

Participants were invited to complete an anonymous questionnaire at the end of the course. Outcomes were measured using validated five-point Likert scale-based ordinal response data (1 = low interest, 5 = high interest) and responses collated. The questionnaire also included open questions on other aspects of the summer school (e.g. how well the course was organised, best/worst aspects plus any additional feedback). Data on some protected characteristics were also collected anonymously.

## Results

Twenty-three applications were received, with 20 students offered places. Seventeen students (8 male, 9 female), representing 8 medical (16 students) and one dental school (1 student) attended the summer school. Six students had just completed their first and 11 their second year of the UK medical/dental curriculum.

All completed the anonymous questionnaire. From this, students' overall impression of the programme was very good, with all, bar one abstention, recommending this type of event for students in the future. Instructors, sessions by clinical academics, venues, social activities, and accommodation all scored highly, with views on the organisation more variable (Fig 1a). In terms of topics covered, students found laboratory safety and pipetting the least interesting, but were much more engaged with histology and digital pathology (Fig 1b). Fourteen students indicated that what they had learned over the week had made them consider a future career as clinical academics, 2 were unsure and one said not as they had already decided to be a clinical

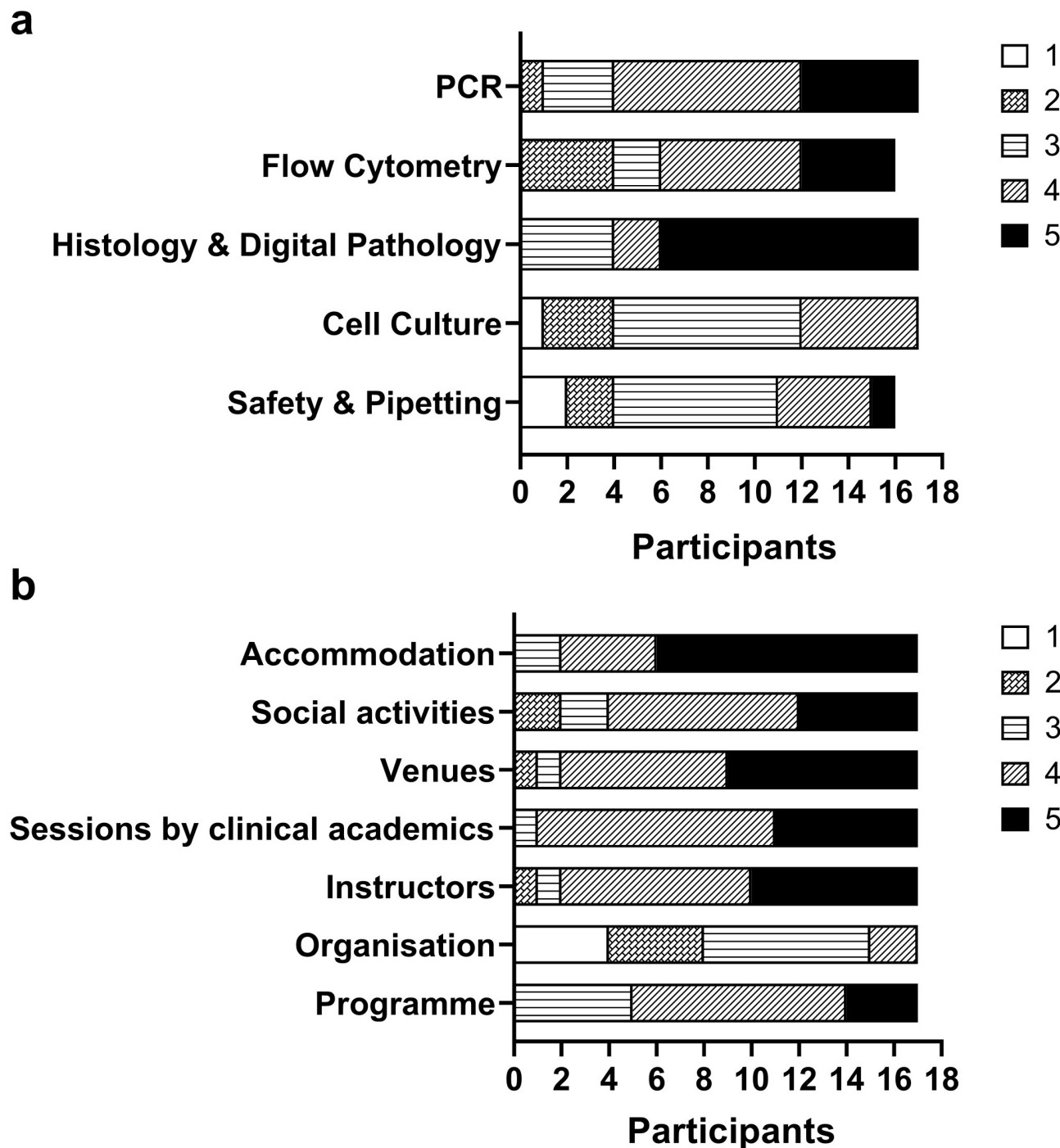

**Fig 1. Student impressions of each aspect of the summer school (a) and their level of interest for each of the topics covered (b).** The number of respondents is shown on the x-axis and their opinions on each of the activities are represented in the bars based on a 5-point Likert scale ranging from 1–5, where 1 = low interest and 5 = high interest.

**Table 1. Qualitative data highlighting the opinion of summer school participants as disclosed in their feedback.**

| Best aspects | Worse aspects | General comments |
| --- | --- | --- |
| Lab-based sessions | Safety and pipetting on first day | Opened my eyes to the opportunities in clinical academia |
| Talks from clinical academics | Technical issues which prevented some activities progressing | I have learnt lots from this one week and has given me push to consider carer as clinical academic |
| Networking with other students and academics | Organisation | I found the talks from the academics thought provoking |
| Learning more about how to get started in a clinical academic career | Too much sitting around | Academic medicine seems like an interesting career option |
| Dinner with academics | Long days/very full on | Developed new skills. Enjoyed the variety of lab skills/activities offered |
| Fantastic learning, bonding & networking experience–and all free! | Lack of information beforehand | Really enjoyed the week/glad I came |

academic before attending the summer school. Participants were invited to include free text comments for any aspect of the course. Relevant comments are shown in Table 1.

## Discussion

The focus of most medical/dental schools nowadays is to train future doctors and dentists to meet the needs of an ageing population with increasingly complex multimorbidity burden. This change in focus has also paralleled the decline in academic medicine. Consequently, medical/dental students are often unaware of academic medicine as a career choice with few pursuing this. Our summer school provided these students with a taste of research and what academic medicine can offer. From their comments it was evident that they developed new competences, enjoyed the variety of laboratory skills and activities that were offered and recommended this type of event for future students.

Interestingly, students were most engaged with histology and digital pathology, which included applying algorithms using the open-source software package QuPath [5] to digital histology images. We did not anticipate this as attracting trainees into pathology has been challenging in recent years. Moves towards digital pathology [6], machine learning [7] and the expansion and routine use of molecular pathology fuelled by the boom in using omics [8] in diagnostic settings may render pathology more exciting and appealing to a generation of digitally literate millennials. The session on laboratory safety and pipetting was deemed least interesting. This was not entirely unexpected as safety in the workplace is often perceived as nagging and boring, and this is how it may have come across for attendees. However, laboratories can be dangerous places and risks associated with using chemicals and pieces of equipment must be conveyed. Hence laboratory safety training is necessary at all levels no matter how tedious it might seem. Some participants found that the days were long with a lot of waiting and there were some technical issues. However, this is often the reality of laboratory research.

Informal sessions with clinical academics of different gender, ethnicity, seniority and across different clinical disciplines provided students with aspirational role models which was highly valued by our participants. These stimulated discussion regarding clinical academic pathways as well as their own career paths and students reported feeling motivated to consider a career in academic medicine.

This was the first time we had organised such an event and we acknowledge there were teething problems. Running this during the summer period coincided with staff holidays which impacted on the organisation and timeliness of the information provided beforehand and was reflected in the comments from attendees. We were also working to a finite budget which limited what we were able to do. We did not restrict who could attend the summer school and some participants were graduate entry. Consequently, the techniques taught were not sufficiently challenging to those with previous life/biomedical science degrees, which was reflected in their feedback. Hence differences in abilities should be considered when running this type of event in the future. There were suggestions from some participants to give each student a mini project to follow for the week or to shadow PhD students in research laboratories. While these are excellent suggestions, they are both time and resource heavy and would inevitably mean restricting this to a much smaller number of students, which would reduce inclusivity and reach for this type of summer school. However, students wishing to pursue this route could engage with academics at their parent medical/dental schools to take this forward, and indeed were encouraged to do so by the organisers.

This summer school has provided a platform for students to explore research options e.g. by participating in research through short projects or considering intercalated degrees at their own universities. Intercalated degrees provide opportunities to gain considerable laboratory skills, and evidence from our own medical school and others shows additional benefits; intercalation improves exam results and helps develops the skills necessary to produce a competitive foundation programme application [9, 10]. However, this adds another year of financial burden on top of an already expensive course, which may pose financial difficulties for some students.

Judging by the feedback from our participants, there appears to be a need for taster programmes like this to introduce students to research and allow them to learn some basic research skills which can be applied in the future. This could inspire more clinicians to follow an academic pathway and coupled with identifying ways of offering financial support for intercalated courses, might offer part of the solution to address the decline in academic medicine as a career choice. We encourage adoption and integration of this type of model for future medical and dental undergraduates.

## Acknowledgments

We are grateful to clinical academics F Clegg, B Elsberger, J Gregory, M Karabayas, A Kiltie, A Lalli, R Soiza and for their time and engagement with participants; A Davidson, A Diack, A Holme, L Lumsden, F Saunders, N Sivamanoharan, D Tosh, L Wight, D Wilkinson, D Wilson, and University of Aberdeen PhD students for technical support and J Forsyth for admin support. The authors gratefully acknowledge the Microscopy and Histology Core Facility and the Iain Fraser Cytometry Centre at the University of Aberdeen for their support and assistance.

## Author Contributions

**Conceptualization:** Valerie Speirs, Rasha Abu-Eid, Phyo Kyaw Myint.

**Data curation:** Valerie Speirs.

**Formal analysis:** Valerie Speirs.

**Funding acquisition:** Valerie Speirs, Rasha Abu-Eid, Phyo Kyaw Myint.

**Investigation:** Valerie Speirs, Rasha Abu-Eid, Phyo Kyaw Myint.

**Methodology:** Valerie Speirs, Rasha Abu-Eid, Phyo Kyaw Myint.

**Project administration:** Valerie Speirs, Rasha Abu-Eid, Phyo Kyaw Myint.

**Writing – original draft:** Valerie Speirs, Rasha Abu-Eid, Phyo Kyaw Myint.

**Writing – review & editing:** Valerie Speirs, Rasha Abu-Eid, Phyo Kyaw Myint.

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
