## [Decision Letter · Decision Letter 0]

3 Oct 2023

PONE-D-23-23768Engaging undergraduate medical and dental students with academic medicine: the Aberdeen INSPIRE summer schoolPLOS ONE

Dear Dr. Speirs,

Thank you for submitting your manuscript to PLOS ONE. After careful consideration, we feel that it has merit but does not fully meet PLOS ONE’s publication criteria as it currently stands. Therefore, we invite you to submit a revised version of the manuscript that addresses the points raised during the review process.

We look forward to receiving your revised manuscript.

Kind regards,

Soham Bandyopadhyay

Academic Editor

PLOS ONE

Additional Editor Comments:

How did students choose what to do?

The figures are not very clear. Could you make them easier to interpret

Reviewers' comments:

Reviewer's Responses to Questions

**Comments to the Author**

1. Is the manuscript technically sound, and do the data support the conclusions?

Reviewer #1: Yes

2. Has the statistical analysis been performed appropriately and rigorously? 

Reviewer #1: Yes

3. Have the authors made all data underlying the findings in their manuscript fully available?

Reviewer #1: Yes

4. Is the manuscript presented in an intelligible fashion and written in standard English?

Reviewer #1: Yes

5. Review Comments to the Author

Reviewer #1: the article was written in a scientific way following the PLOS ONE guidelines, including introduction, research idea, results and discussion. the results and conclusion met the objectives of the study.

6. PLOS authors have the option to publish the peer review history of their article (what does this mean?). If published, this will include your full peer review and any attached files.

Reviewer #1: No

---

## [Author Response · Author response to Decision Letter 0]

5 Oct 2023

Thank you for reviewing our manuscript (PONE-D-23-23768).

In response to the journal comments:

This has been done.

Checked and all correct.

In response to Additional Editor Comments:

How did students choose what to do?

The programme was fixed with all students completing the same activities. This sentence has been added (line 68).

The figures are not very clear. Could you make them easier to interpret

Apologies for the lack of clarity. We have named the x-axis on both Fig 1a and b to reflect that each of the numbers refer to the participants. The Likert scoring has been clarified (line 72) and the legend for Fig 1 has been expanded with an additional sentence “The number of respondents is shown on the x-axis and their opinions on each of the activities are represented in the bars based on a 5-point Likert scale ranging from 1- 5, where 1 = low interest and 5 = high interest.”

Reviewers' comments:

No comments to address.

---

## [Editor Report · Decision Letter 1]

17 Oct 2023

Engaging undergraduate medical and dental students with academic medicine: the Aberdeen INSPIRE summer school

PONE-D-23-23768R1

Dear Dr. Speirs,

We’re pleased to inform you that your manuscript has been judged scientifically suitable for publication and will be formally accepted for publication once it meets all outstanding technical requirements.

Kind regards,

Soham Bandyopadhyay

Academic Editor

PLOS ONE
---

## [Editor Report · Acceptance letter]

27 Oct 2023

PONE-D-23-23768R1 

Engaging undergraduate medical and dental students with academic medicine: the Aberdeen INSPIRE summer school 

Dear Dr. Speirs:

I'm pleased to inform you that your manuscript has been deemed suitable for publication in PLOS ONE. Congratulations! Your manuscript is now with our production department. 

Kind regards, 

on behalf of

Dr. Soham Bandyopadhyay 

Academic Editor

PLOS ONE